# Diagnosing a Patient with Erdheim-Chester Disease during the COVID-19 Pandemic

**DOI:** 10.3390/medicina57101001

**Published:** 2021-09-22

**Authors:** Georgia Kaiafa, Dimitrios Pilalas, Triantafyllia Koletsa, Stylianos Daios, Georgios Arsos, Adam Hatzidakis, Adonis Protopapas, Kostas Stamatopoulos, Christos Savopoulos

**Affiliations:** 1First Propedeutic Department of Internal Medicine, AHEPA University Hospital, Medical School, Aristotle University of Thessaloniki, 54636 Thessaloniki, Greece; pilalas_jim@hotmail.com (D.P.); stylianoschrys.daios@gmail.com (S.D.); adoprot@hotmail.com (A.P.); chrisavopoulos@gmail.com (C.S.); 2Department of Pathology, Medical School, Aristotle University of Thessaloniki, 54124 Thessaloniki, Greece; tkoletsa@auth.gr; 33rd Department of Nuclear Medicine, Papageorgiou Hospital, Aristotle University of Thessaloniki, 56403 Thessaloniki, Greece; garsos@auth.gr; 4Radiology Department, AHEPA University Hospital, Medical School, Aristotle University of Thessaloniki, 54636 Thessaloniki, Greece; adamhatz@hotmail.com; 5Institute of Applied Biosciences, Center for Research and Technology Hellas, 57001 Thessaloniki, Greece; kostas.stamatopoulos@certh.gr

**Keywords:** Erdheim-Chester disease, COVID-19, case report

## Abstract

*Background*: Erdheim-Chester disease (ECD) is a rare hematopoietic neoplasm of histiocytic origin characterized by an insidious course. The coronavirus disease 2019 (COVID-19) pandemic has put an enormous strain on healthcare systems worldwide both directly and indirectly, resulting in the disruption of healthcare services to prevent, diagnose and manage non-COVID-19 disease. *Case Presentation*: We describe the case of a 58-year-old male patient with sporadic episodes of self-resolving mild fever and anemia of chronic disease with onset two years before the current presentation. Positron emission/computed tomography scan revealed the presence of moderately hypermetabolic perirenal tissue masses. In order to achieve diagnosis, repeated perirenal tissue biopsies were performed, and the diagnostic evaluation was complicated by the strain put on the healthcare system by the COVID-19 pandemic. The patient contracted SARS-CoV-2 and required hospitalization, but recovered fully. No further ECD target organ involvement was documented. Treatment options were presented, but the patient chose to defer treatment for ECD. *Conclusion*: A high index of suspicion and multidisciplinary team collaboration is paramount to achieve diagnosis in rare conditions such as ECD. Disruptions in healthcare services in the pandemic milieu may disproportionately affect people with rare diseases and further study and effort is required to better meet their needs in the pandemic setting.

## 1. Introduction

Erdheim-Chester disease (ECD) is a rare clonal histiocytic disorder, reclassified as histiocytic neoplasm in the 2016 WHO classification, most commonly characterized by an insidious course with cumulative tissue infiltration by foamy CD68+CD1a- histiocytes; the progressive multisystem disease can be fatal if untreated [1,2]. Histiocytoses are a clinically heterogeneous group of rare disorders characterized by infiltration of any tissue with protean manifestations [3]. The histology and manifestations may not always allow to differentiate between different entities, while the molecular underlying defects may be common (e.g. ECD, cases of Rosai-Dorfman disease and juvenile xanthogranuloma) [3]. ECD predominantly affects men (ratio 2.4:1) and the median age at diagnosis is 55 years [3]. The interval between symptom onset and diagnosis has decreased in recent years and was estimated to be on average 2.7 years in a case series comprising 261 patients out of approximately 1500 ever diagnosed with the disease globally [1].

The coronavirus disease 2019 (COVID-19) pandemic has put an enormous strain on healthcare systems worldwide both directly and indirectly, resulting in the disruption of healthcare services to prevent, diagnose and manage non-COVID-19 disease [4]. Patients with rare diseases were found in a particularly vulnerable position as frequently diagnosis and management require a concerted effort unattainable in the pandemic milieu [5]. Furthermore, data with regard to the implications of COVID-19 in the context of these diseases are rather unavailable [5].

In our report, we describe the diagnostic process in a patient who was ultimately diagnosed with Erdheim-Chester disease (ECD) during the COVID-19 pandemic. The patient was also infected with COVID-19, resulting in an additional delay of 3 months prior to proving ECD diagnosis. 

## 2. Case Report 

A 58-year-old male patient presented with sporadic self-resolving episodes of mild fever (up to five episodes per month with duration of several hours) and anemia of chronic disease with onset two years before current presentation. Patient history was notable for Type 2 diabetes mellitus (HbA1c: 6.8% on metformin) and dyslipidemia treated with atorvastatin.

Following the onset of fever episodes, a workup was undertaken: clinical examination was normal, and laboratory results revealed a mild borderline microcytic normochromic anemia (Table 1). There was no evidence of hemolysis while the iron parameters were compatible with anemia of chronic disease (ferritin 135 ng/mL along with low transferrin saturation attributable to functional iron deficiency anemia). C-reactive protein (CRP) was elevated (1.55 mg/dL) (0.0–0.8 mg/dL), but erythrocyte sedimentation rate was considered normal, adjusting for age and taking into account the level of anemia. The peripheral blood and the bone marrow aspirate smear did not show any dysplastic features such as dyserythropoiesis, neutrophil dysplasia, micromegakaryocytes, nor peripheral blood monocytes, and the myeloblasts constituted less than 2% of the nucleated bone marrow cells. Common infectious and autoimmune conditions presenting as fever of unknown origin were excluded, while gastrointestinal endoscopy was not contributive [6]. On abdominal computed tomography, perirenal fat and fascia infiltration were noted. An 18F-fluorodeoxyglucose (18F-FDG) positron emission tomography/computed tomography (PET/CT) scan demonstrated moderately increased radiotracer uptake of perirenal tissue (Figure 1). A CT-guided perirenal tissue biopsy revealed inflammatory reaction and fibrosis without features of a specific disease. Our diagnostic hypothesis of lymphoma was not corroborated by the perirenal tissue and bone marrow biopsy.

The decision was made to follow the patient with the intention to repeat the biopsy. However, due to the disruption of outpatient clinics in public hospitals during the first wave of the COVID-19 pandemic, the patient was reevaluated approximately one year after the first biopsy. The symptoms and mild anemia persisted. PET-CT was repeated without new findings. Immunoglobulin G4-related disease (IgG4-RD) was considered, and a borderline elevated serum IgG4 of 155 mg/dL (8–140 mg/dL) was detected. Perirenal tissue biopsy was repeated, and the morphologic findings were compatible with IgG4-RD, but immunohistochemically, there was no increase in the number of IgG4+ plasma cells nor an abnormal IgG4+/IgG+ ratio. 

The case was reviewed, and the diagnosis of ECD was evoked. In this context, histological slides from previous biopsies were reviewed and confirmed the dense lymphoplasmacytic infiltrate, predominantly plasmacytic, in a background of fibrosis, with scarce histiocytes. Notably, there was no bone marrow infiltration. Taking under consideration that ECD diagnosis may be established after consecutive biopsies, there was a clinical suspicion of a non-representative biopsy, and a new biopsy was scheduled. At admission, which coincided with the plateau of the second wave of the COVID 19 pandemic, the patient was febrile without any signs of distress (blood pressure:140/85 mmHg, heart rate 94/min, respiratory rate of 20/min), and a molecular SARS-CoV-2 test on a nasopharyngeal sample confirmed infection with SARS-CoV-2. Arterial blood gases indicated the patient was mildly hypoxemic pH 7.41, PaO_2_: 68 mmHg, HCO_3_^-^:23 mmol/l, PaCO_2_: 32 mmHg), while laboratory results were notable for an elevated CRP value to 1.6 mg/dL consistent with previous findings. A chest CT scan demonstrated bilateral ground-glass opacifications with peripheral distribution affecting the right lower lobe predominantly (Figure 2).

The patient needed oxygen supplementation with nasal cannula and was treated according to the institutional protocol at the time with azithromycin 500 mg once daily for 3 days, dexamethasone 6 mg once daily for 8 days and intermediate dose prophylactic anticoagulation. He improved gradually and was discharged in stable condition without persisting symptoms. It should be noted that a vaccination against COVID-19 was not yet available at the time.

The perirenal tissue biopsy was performed 2 months later under ultrasound guidance. Histological examination revealed a few histiocytes with abundant foamy (xanthomatous) cytoplasm and small nuclei, either single or in small clusters, into a fibrotic stroma. Multinucleated histiocytes were rarely identified. Immunohistochemically, these cells were positive for CD68, fascin and cyclinD1, and negative for S100, CD1a, ALK and tryptase. The findings were consistent with ECD upon correlating pathological with clinical and especially radiological features (“hairy kidney” appearance). Targeted next-generation sequencing demonstrated the presence of mutation V600E of BRAF on gene exon 15.

Following diagnosis confirmation, the workup was extended and excluded further target organ involvement (heart, brain and pituitary magnetic resonance imaging, and endocrine function testing) [7]. A bone marrow biopsy was repeated to exclude a concomitant myeloid neoplasm [7]. Treatment options were presented to the patient, but the patient chose to defer treatment.

## 3. Discussion

We present a patient with intermittent fever and anemia of chronic disease diagnosed with ECD during the COVID-19 pandemic who also experienced COVID-19 disease while under investigation. 

The clinical presentation of ECD is highly variable and depends on the sites of histiocyte infiltration. Typical manifestations include long bone osteosclerosis (80–95%), “hairy kidney” sign (63%), coated aorta (40%) and central nervous system involvement (38%), right atrium pseudotumor (36%), diabetes insipidus (25%) and xanthelasma (22%) [1,8]. Both fever and anemia have been previously reported at the presentation of ECD [9,10], but these findings are neither common nor specific [1,11]. The aforementioned symptoms combined with the perirenal tissue infiltration prompted us to exclude lymphoma. 

Following up on the finding of positive IgG4 serology, the diagnosis of IgG4-RD was pursued. IgG4-RD may present with tumor-like retroperitoneal lesions, and involvement is frequently multisystem with the possibility to affect virtually every organ [12]. IgG4+ plasma cells predominate in dense lymphoplasmacytic infiltrations, which are characteristic of the disease, along with storiform fibrosis, tissue eosinophilia and obliterative phlebitis [12]. However, retroperitoneal fibrosis in Ig4-RD usually has a periaortic and periiliac pattern, and in our case, IgG4 immunostain normal values of IgG4+ plasma cells and a normal IgG4+/IgG ratio oriented the diagnosis away from IgG4-RD [13]. 

The exclusion of the aforementioned diagnoses combined with the presence of the “hairy kidney” sign raised the index of suspicion for ECD, despite the absence of characteristic symmetric osteosclerotic long bone lesions in a dedicated head-to-toe PET-CT. In ECD, tissue is infiltrated by histiocytes with foamy (xanthomatous) cytoplasm and small, round nuclei embedded in extensive fibrosis. Touton giant cells and chronic inflammation are also frequently encountered. As the histological appearance is not specific, it is common for ECD to be confused with a reactive process [14], leading to a delay in diagnosis. Perirenal tissue biopsy was repeated, and large xanthomatous cells were observed only in the last biopsy specimen. ECD histiocytes are immunoreactive for CD68, CD163, CD14, factor XIIIa, fascin and CD45, whereas CD1a and CD207 (langerin) are negative. S100 is usually negative but may appear positive in some cases [8]. Regarding the molecular features, BRAF V600E is the most common mutation, detected in more than 50% of the cases [7,8] followed by MAP2K1, KRAS, NRAS, Ρ13ΚCA and ARAF mutations [7]. ECD shares features with other systemic histiocytic disorders such as Rosai-Dorfman disease (RDD) and Langerhans cell histiocytosis (LCH), and these entities should be considered in the differential diagnosis [11]. Diagnosing ECD is challenging and requires a multidisciplinary approach. 

ECD staging should exclude heart, central nervous system and endocrine system involvement with implications for treatment. Hypothyroidism is highly prevalent in ECD patients and may also constitute an adverse event of ECD treatment with interferon-alpha [15,16]. In our case, the primary involvement was the perirenal tissue infiltration with a risk for hydronephrosis and interferon-alpha was proposed as first line treatment. Targeted therapies are preferred in high-risk disease and have revolutionized ECD treatment. Their efficacy is dependent on the molecular mechanism underlying disease and, in our case, vemurafenib, a BRAF-inhibitor, would be an alternative given the presence of mutation V600E of the BRAF gene. However, the risk of adverse events and the risk of reactivation after treatment discontinuation probably weighed in the decision of our patient to defer treatment.

The COVID-19 pandemic impact resulted in a 63% and 90% decline in the diagnosis of rare diseases in March and April 2020, respectively, compared to the previous year in the rare disease healthcare network in Campania, Italy [17]. A survey embedded in the same study identified difficulty in establishing the diagnosis, follow-up, and the need for home therapy among the most critical issues raised by physicians of the network during the COVID-19 pandemic [17]. Similar considerations have been documented among patients with rare diseases in various settings [18,19]. In our case, the diagnostic challenges ECD poses were exacerbated by COVID-19 and necessitated almost 2.5 years to reach a diagnosis. 

Of note is that 95% of patients diagnosed with a rare disease in this period in the Campania rare disease healthcare network had also been diagnosed with COVID-19 [17]. The COVID-19 disease course in our patient necessitated supplemental oxygen with nasal cannula but was not complicated further. To our knowledge, we report the first case of COVID-19 in a patient with ECD.

## 4. Conclusions

A high index of suspicion and multidisciplinary team collaboration is paramount to achieve diagnosis in rare conditions such as ECD. Disruptions in healthcare services in the pandemic milieu may disproportionately affect people with rare diseases, and further study and effort are required to better meet their needs in the pandemic setting.

## Figures and Tables

**Figure 1 medicina-57-01001-f001:**
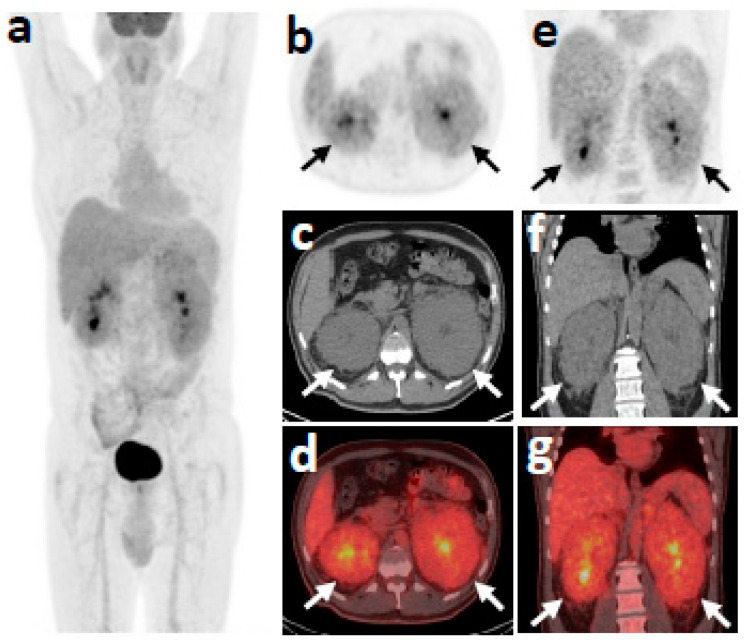
18F-FDG-PET/CT skull base to mid-thigh scan at first patient presentation. Blood glucose level at injection time 165 mg/dL. (**a**) Maximum intensity projection (MIP); (**b**–**g**), axial and coronal, PET, CT and fused sections at the level of kidneys, respectively. Bulky, moderately hypermetabolic (SUVmax 3.8) perirenal soft tissue masses (arrows) surrounding both kidneys. The pyelocalyceal system of the kidneys is observed in the center of the masses. No other hypermetabolic focus was detected.

**Figure 2 medicina-57-01001-f002:**
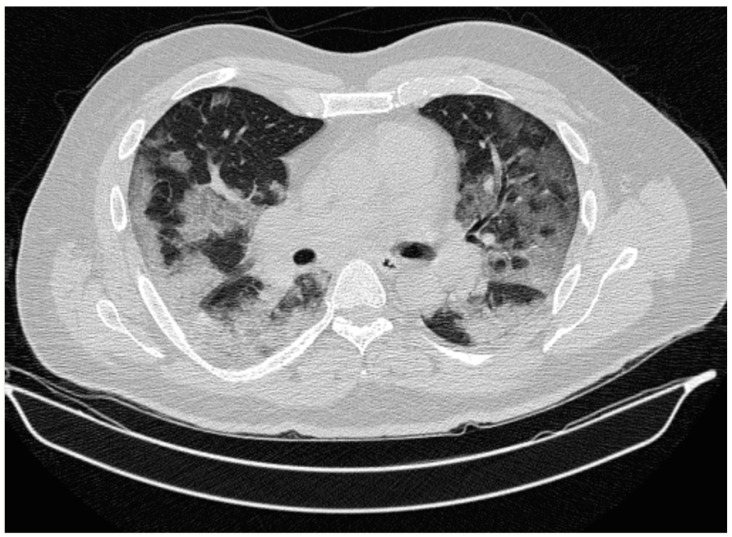
Chest CT scan in lung window: Bilateral ground-glass opacifications with peripheral distribution affecting the right lower lobe predominantly.

**Table 1 medicina-57-01001-t001:** Laboratory values.

Laboratory Parameter	Patient Value on Admission	Reference Value
Hemoglobin	11.9 g/dL	14.0–18.0 g/dL
Mean Corpuscular Volume	79 fL	80.0–99.0 fL
Mean Corpuscular Hemoglobin	28.3 pg	27.0–32.0 pg
Reticulocytes	0.9%	0–2%
White Blood Cells	6.59 Κ/μL	3.8–10.5 K/μL
Neutrophils	3.9 K/μL	1.6–6.5 K/μL
Monocytes	0.45 Κ/μL	0.2–1.0 Κ/μL
Platelets	398 K/μL	150–450 Κ/μL
Ferritin	135 ng/mL	0–400 ng/mL
Urea	34 mg/dL	10–50 mg/dL
Creatinine	1.02 mg/dL	0.50–1.20 mg/dL
AST	11 U/L	0–38 U/L
ALT	7 U/L	0–40 U/L
Erythrocyte Sedimentation rate	35 mm	0–20 mm
CRP	1.55 mg/dL	0.0–0.8 mg/dL
HbA1c	6.8%	4–6%
TSH	3.01 μIU/mL	0.27–4.20 μIU/mL
FT3	3.6 pmol/L	3.1–6.8 pmol/L
FT4	16.4 pmol/L	12.0–22.0 pmol/L

## Data Availability

All relevant data are included in the manuscript.

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
