# Peer review of "Diagnosing a Patient with Erdheim-Chester Disease during the COVID-19 Pandemic"

_medicina, 2021, doi:10.3390/medicina57101001_

Round 1
Reviewer 1 Report
Interesting manuscript on rare disease diagnosis in the setting of the SARS-cov2 pandemy
the introduction could be improved with a few sentences replacing erdheim-chester disease in the broader spectrum of other histiocytoses. in line with thi sthe authors could add to the references the 2021 Seminar Lancet review on the subject by Emile et al. the age of pts as well as sex ratio, association with other histiocytoses should also be mentionned
it would also be important to give more details on the blood formula and see if there was any argument for CMML or MDS.
even if in this case was not treated with targeted therapy due to the absence of access the importance and efficacy of such drugs should be emphasized a bit more
did this patient had the Sars-Cov2 vaccine ? which one ? please comment and edit accordingly
Author Response
Dear Reviewer #1,
We were very pleased to receive the evaluation of our manuscript and we would like to thank you for your insightful comments. We have addressed all the concerns raised in a revised version of the manuscript as requested. All changes in the manuscript are highlighted in yellow format (resubmitted file) along with the changes requested by reviewer #2.
Please find below our response, point-by-point, to the comments provided by the Reviewers.
Reviewer #1:
Interesting manuscript on rare disease diagnosis in the setting of the SARS-cov2 pandemy
-We would like to thank you for your kind comments.
the introduction could be improved with a few sentences replacing erdheim-chester disease in the broader spectrum of other histiocytoses. in line with this the authors could add to the references the 2021 Seminar Lancet review on the subject by Emile et al. the age of pts as well as sex ratio, association with other histiocytoses should also be mentioned
-In the first paragraph of the manuscript we modified the text that follows:”
Histiocytoses are a clinically heterogeneous group of rare disorders characterized by infiltration of any tissue with protean manifestations. The histology and manifestations may not always allow to differentiate between different entities, while the molecular underlying defects may be common (e.g. ECD, cases of Rosai-Dorfman disease and juvenile xanthogranuloma). ECD predominantly affects men (ratio 2.4:1) and the median age at diagnosis is 55 years.
It would also be important to give more details on the blood formula and see if there was any argument for CMML or MDS.
- We have added the following details in the presentation of the case: “The peripheral blood and the bone marrow aspirate smear did not show any dysplastic features such as dyserythropoiesis, neutrophil dysplasia, micromegakaryocytes, nor peripheral blood monocytes and the myeloblasts constituted less than 2% of the nucleated bone marrow cells.” Exact values values are presented in Table 1.
even if in this case was not treated with targeted therapy due to the absence of access the importance and efficacy of such drugs should be emphasized a bit more
-We have updated the text as meanwhile access to targeted therapy was made possible, but the patient finally chose to defer treatment despite our recommendation to initiate treatment with one of the alternative possibilities. The following phrase was added in the case presentation: Treatment options were presented to the patient, but the patient chose to defer treatment. The following paragraph was added in the discussion section:”
We added the following paragraph regarding the importance and efficacy of mentioned targeted therapy: “In our case, the primary involvement was the perirenal tissue infiltration with a risk for hydronephrosis and interferon-alpha was proposed as first line treatment. Targeted therapies are preferred in high risk disease and have revolutionized ECD treatment. Their efficacy is dependent on the molecular mechanism underlying disease and in our case vemurafenib, a BRAF-inhibitor would be an alternative given the presence of mutation V600E of BRAF gene. However, the risk of adverse events and the risk of reactivation after treatment discontinuation probably weighed in the decision of our patient to defer treatment.”
did this patient had the Sars-Cov2 vaccine ? which one ? please comment and edit accordingly
-The patient had not been vaccinated against Sars-CoV-2 because the vaccine was unavailable at the time. We have inserted a relevant passage.
Finally in order to reflect the updated version of the manuscript we modified the abstract as follows (These are only the modifications included, the whole abstract is written inside the main manuscript) :
Background section: Erdheim-Chester disease (ECD) is a rare haematopoietic neoplasm of histiocytic origin characterized by an insidious course.
Case presentation section: We describe the case of a 58-year-old male patient with episodes of sporadic self-resolving mild fever and anemia of chronic disease with onset two years before the current presentation. Positron emission/computed tomography scan revealed the presence of moderately hypermetabolic perirenal tissue masses. In order to achieve diagnosis, repeated perirenal tissue biopsies were performed and the diagnostic evaluation was complicated by the strain put on the healthcare system by the COVID-19 pandemic. The patient contracted SARS-CoV-2 and required hospitalization, but recovered fully. No further ECD target organ involvement was documented. Treatment options were presented, but the patient chose to defer treatment for ECD.
We declare there is no conflict of interest in our submission and all authors have
approved the final version of the manuscript.
We hope the revised and improved version of our manuscript will be deemed suitable for publication in Medicina Journal.
We look forward to hearing from you.
Reviewer 2 Report
Authors describe an interesting case of Erdheim-Chester disease in a middle aged male who presented with chronic, non-specific symptoms, delayed in the setting of COVID-19 pandemic. I find this case to be well written and flows logically but have some concerns and suggestions for improvement
Introduction
- Authors describe ECD as a ‘Erdheim-Chester disease (ECD) is a rare clonal histiocytic disorder’. It is also important to note that it has been reclassified as hematopoietic neoplasm of histiocytic origin (making it a blood cancer) by the WHO in 2016. Please include and cite PMID: 26980727
Case report
- Authors report that the patient had type 2 DM and hypertension but do not list the glycemic control (Hba1c) or blood pressure values and medications taken by the patient. Please elaborate.
- CRP elevation is mentioned on presentation without listing actual values (1.6 is listed as a subsequent value). This is also the case with IgG4. Please clarify. In general, I recommend presenting relevant labs in a tabular format with normal ranges.
- Authors state ‘Common infectious and autoimmune diseases were excluded’. What were the infections and autoimmune disease excluded? Authors need to elaborate.
- On COVID-19 admission, the patient’s hemodynamics need to be describe especially since he required nasal cannula oxygen. Please elaborate.
- Authors mention ‘The biopsy was performed two months later’. What was the site of the biopsy? Please mention.
Discussion
- Authors have relied on one case series and case reports, but that other relevant case series should also be discussed to provide a comprehensive picture. PMID: 28553668
- For this sentence ‘S100 is usually negative but may appear positive in some cases.’ Please provide a suitable reference.
- Given that the patient was started on interferon therapy (or planned for it) and his history of normocytic anemia, was hypothyroidism excluded? Hypothyroidism is a highly prevalent and underdiagnosed condition in ECD. I recommend that this information is presented and discussed. Please review and refer to PMID: 33119105 in the discussion.
- Other recommended screenings include endocrine, neurological, brain & pituitary imaging. If they were performed it should be listed or else a plan to execute these should be elaborated to inform readers of best practices. I recommend including the above mentioned discussion points described in these articles
https://ashpublications.org/blood/article/135/22/1929/452713/Erdheim-Chester-disease-consensus-recommendations
https://academic.oup.com/jcem/article/101/1/305/2806902
https://www.mdpi.com/2072-6694/13/16/4126
Author Response
Dear Reviewer #2,
We were very pleased to receive the evaluation of our manuscript and we would like to thank you for your insightful comments. We have addressed all the concerns raised in a revised version of the manuscript as requested. All changes in the manuscript are highlighted in yellow format (resubmitted file) along with the changes requested by reviewer #1.
Please find below our response, point-by-point, to the comments provided by the Reviewers.
Reviewer #2:
Authors describe an interesting case of Erdheim-Chester disease in a middle aged male who presented with chronic, non-specific symptoms, delayed in the setting of COVID-19 pandemic. I find this case to be well written and flows logically but have some concerns and suggestions for improvement
-We would like to thank you for your kind comments.
Introduction
Authors describe ECD as a ‘Erdheim-Chester disease (ECD) is a rare clonal histiocytic disorder’. It is also important to note that it has been reclassified as hematopoietic neoplasm of histiocytic origin (making it a blood cancer) by the WHO in 2016. Please include and cite PMID: 26980727
- We modified the beginning of the introduction as follows:”Erdheim-Chester disease (ECD) is a rare clonal histiocytic disorder, reclassified as histiocytic neoplasm in the 2016 WHO classification, most commonly characterized by an insidious course with cumulative tissue infiltration by foamy CD68+CD1a- histiocytes; the progressive multisystem disease can be fatal if untreated.”
Case report
Authors report that the patient had type 2 DM and hypertension but do not list the glycemic control (Hba1c) or blood pressure values and medications taken by the patient. Please elaborate.
-The patient history was notable for type 2 DM and dyslipidemia. We have added the HbA1c and the treatment the patient was on.
CRP elevation is mentioned on presentation without listing actual values (1.6 is listed as a subsequent value). This is also the case with IgG4. Please clarify. In general, I recommend presenting relevant labs in a tabular format with normal ranges.
-CRP actual values were added (mg/dl). The subsequent value was reported for COVID-19 Infection. IgG4 actual values were also added (mg/dl) along with the reference values. A table containing the most relevant laboratory parameters was created (Table 1).
Authors state ‘Common infectious and autoimmune diseases were excluded’. What were the infections and autoimmune disease excluded? Authors need to elaborate.
-In order to increase clarity we modified the phrase as follows adding a reference indicating our approach to fever of unknown origin:” Common infectious and autoimmune conditions presenting as fever of unknown origin were excluded, while gastrointestinal endoscopy was not contributive [PMID: 26093175, DOI: 10.1016/j.amjmed.2015.06.001].”
On COVID-19 admission, the patient’s hemodynamics need to be describe especially since he required nasal cannula oxygen. Please elaborate.
- We modified the COVID-19 part as follows to make it more clear and educating: “At admission, which coincided with the plateau of the second wave of the COVID 19 pandemic, the patient was febrile without any signs of distress (blood pressure:140/85mmHg, heart rate 94/min, respiratory rate of 20/min), and a molecular SARS-CoV-2 test on a nasopharyngeal sample confirmed infection with SARS-CoV-2. Arterial blood gases indicated the patient was mildly hypoxemic pH 7.41, PaO2: 68mmHg, HCO3-:23mmol/l, PaCO2: 32mmHg), while laboratory results were notable for an elevated CRP value to 1.6mg/dl (0.0-0.8 mg/dl) consistent with previous findings. A chest CT scan demonstrated bilateral ground-glass opacifications with peripheral distribution affecting the right lower lobe predominantly (Figure 2).
The patient necessitated oxygen supplementation with nasal cannula and was treated according to the institutional protocol at the time with azithromycin 500mg once daily for 3 days, dexamethasone 6mg once daily for 8 days and intermediate dose prophylactic anticoagulation. He improved gradually and was discharged in stable condition without persisting symptoms. It should be noted that a vaccination against COVID-19 was not yet available at the time.”
In the discussion we have modified a relevant phrase as follows: The COVID-19 disease course in our patient necessitated supplemental oxygen with nasal cannula but was not complicated further.
Authors mention ‘The biopsy was performed two months later’. What was the site of the biopsy? Please mention.
-We included the phrase “Perirenal tissue biopsy” as recommended.
Discussion
Authors have relied on one case series and case reports, but that other relevant case series should also be discussed to provide a comprehensive picture. PMID: 28553668
-PMID: 28553668 was added and discussed appropriately.
For this sentence ‘S100 is usually negative but may appear positive in some cases.’ Please provide a suitable reference.
- We provided a suitable reference (PMID: 28553668)
Given that the patient was started on interferon therapy (or planned for it) and his history of normocytic anemia, was hypothyroidism excluded? Hypothyroidism is a highly prevalent and underdiagnosed condition in ECD. I recommend that this information is presented and discussed. Please review and refer to PMID: 33119105 in the discussion.
We added the following text in the discussion section: “Hypothyroidism is highly prevalent in ECD patients and may also constitute an adverse event of ECD treatment with interferon-alpha. In our case, the primary involvement was the perirenal tissue infiltration with a risk for hydronephrosis and interferon-alpha was proposed as first line treatment.”
Other recommended screenings include endocrine, neurological, brain & pituitary imaging. If they were performed it should be listed or else a plan to execute these should be elaborated to inform readers of best practices. I recommend including the above mentioned discussion points described in these articles
-We added the following sentence according to your recommendation in the case report section “Following diagnosis confirmation, the workup was extended to exclude further target organ involvement (heart, brain and pituitary magnetic resonance imaging, endocrine function testing).”
-We also added the following sentence according to your recommendation in the discussion section “ECD staging should exclude heart, central nervous system and endocrine system involvement with implications for treatment.”
https://ashpublications.org/blood/article/135/22/1929/452713/Erdheim-Chester-disease-consensus-recommendations
https://academic.oup.com/jcem/article/101/1/305/2806902
https://www.mdpi.com/2072-6694/13/16/4126
- The relevant references were added and discussed inside the text.
Finally in order to reflect the updated version of the manuscript we modified the abstract as follows (These are only the modifications included, the whole abstract is written inside the main manuscript) :
Background section: Erdheim-Chester disease (ECD) is a rare haematopoietic neoplasm of histiocytic origin characterized by an insidious course.
Case presentation section: We describe the case of a 58-year-old male patient with episodes of sporadic self-resolving mild fever and anemia of chronic disease with onset two years before the current presentation. Positron emission/computed tomography scan revealed the presence of moderately hypermetabolic perirenal tissue masses. In order to achieve diagnosis, repeated perirenal tissue biopsies were performed and the diagnostic evaluation was complicated by the strain put on the healthcare system by the COVID-19 pandemic. The patient contracted SARS-CoV-2 and required hospitalization, but recovered fully. No further ECD target organ involvement was documented. Treatment options were presented, but the patient chose to defer treatment for ECD.
We declare there is no conflict of interest in our submission and all authors have
approved the final version of the manuscript.
We hope the revised and improved version of our manuscript will be deemed suitable for publication in Medicina Journal.
We look forward to hearing from you.
Round 2
Reviewer 2 Report
Acceptable as is.